# Identifying Sheep Activity from Tri-Axial Acceleration Signals Using a Moving Window Classification Model

**Jamie Barwick** [1,*,†]**, David William Lamb** [1,2]**, Robin Dobos** [1,3]**, Mitchell Welch** [1]**, Derek Schneider** [1] **and Mark Trotter** [1,‡]

[1]   Precision Agriculture Research Group, University of New England, Armidale, NSW 2351, Australia;
     dave.lamb@foodagility.com (D.W.L.); robin.dobos@dpi.nsw.gov.au (R.D.); mwelch8@une.edu.au (M.W.);
     dschnei5@une.edu.au (D.S.); m.trotter@cqu.edu.au (M.T.)
[2]   Food Agility CRC, University of New England, Armidale, NSW 2351, Australia
[3]   NSW Department of Primary Industries, Livestock Industries Centre, University of New England,
     Armidale, NSW 2351, Australia
*   Correspondence: jbarwic2@une.edu.au
†   Current address: Sheep CRC, University of New England, Armidale, NSW 2351, Australia.
‡   Current address: Central Queensland University, CQIRP, Rockhampton, QLD 4702, Australia.

**Abstract:** Behaviour is a useful indicator of an individual animal's overall wellbeing. There is widespread agreement that measuring and monitoring individual behaviour autonomously can provide valuable opportunities to trigger and refine on-farm management decisions. Conventionally, this has required visual observation of animals across a set time period. Technological advancements, such as animal-borne accelerometers, are offering 24/7 monitoring capability. Accelerometers have been used in research to quantify animal behaviours for a number of years. Now, technology and software developers, and more recently decision support platform providers, are integrating to offer commercial solutions for the extensive livestock industries. For these systems to function commercially, data must be captured, processed and analysed in sync with data acquisition. Practically, this requires a continuous stream of data or a duty cycled data segment and, from an analytics perspective, the application of moving window algorithms to derive the required classification. The aim of this study was to evaluate the application of a 'clean state' moving window behaviour state classification algorithm applied to 3, 5 and 10 second duration segments of data (including behaviour transitions), to categorise data emanating from collar, leg and ear mounted accelerometers on five Merino ewes. The model was successful at categorising grazing, standing, walking and lying behaviour classes with varying sensitivity, and no significant difference in model accuracy was observed between the three moving window lengths. The accuracy in identifying behaviour classes was highest for the ear-mounted sensor (86%–95%), followed by the collar-mounted sensor (67%–88%) and leg-mounted sensor (48%–94%). Between-sheep variations in classification accuracy confirm the sensor orientation is an important source of variation in all deployment modes. This research suggests a moving window classifier is capable of segregating continuous accelerometer signals into exclusive behaviour classes and may provide an appropriate data processing framework for commercial deployments.

**Keywords:** accelerometer; activity; behaviour; livestock; monitoring; sheep

## 1. Introduction

On-animal-sensors capable of measuring individual animal behaviour and location have long been considered a potentially transformative technology for extensive livestock grazing enterprises.

Such sensors have been proposed to alleviate many of the labour and cost challenges associated with monitoring livestock health [1]. One particular class of sensor, accelerometers, offer the capability to monitor changes in physical behaviour purely on the basis of movement and orientation. Such information could be used to create animal health and welfare indicators on the basis of deviations in activity patterns from baseline levels [2,3]. The capability of accelerometers to measure posture and activity states has been well established in ruminants (cattle [4–8], goats [9–11] and sheep [3,12–17]).

In a commercial scenario, and especially under extensive grazing conditions, sensors must be able to monitor animal behaviour in near-real-time, and more or less continuously '24/7' (or via a time segmented duty cycle program) if they are to enable accurate and timely management decisions aimed at optimising animal performance and welfare [4]. The advantages and possible uses of a real-time behaviour classification system for extensive livestock production have been widely postulated [18]. However, research to date has focused on applying it to commercial intensive dairy and beef feedlot sectors (for example, the IceTag3D™, REDI, SCR/Alflex and CowManager Sensor systems). Limited research has been conducted on the extensive sheep grazing industry, and there has been little consideration given to optimising data processing and analysis protocols.

The task of developing a near-real-time classification model on the basis of live sensor data is inherently more difficult than developing an off-line classifier which is effectively post-processing of historical data. Inherent constraints and challenges include: (i) correlated time-dependent measurements; (ii) transitional states; and (iii) rapid signal processing requirement with minimal computational energy and storage overhead [8]. Many studies have focused on the classification of data where the start and end times of each behaviour state are known and the use of epochs, which groups data over a predetermined time period, is a common analysis step [6–8,12,19].

These epoch values (derived from calculated metrics) are assumed to be representative of the estimated intensity of activities measured during the set time period [20]. Selecting a short epoch may be suitable if activity is accumulated in a number of short bouts, that is, animals transition between activity states rapidly. A longer epoch has the advantage of offering data-smoothing through time averaging, while the disadvantage is that a higher proportion may contain a mixture of two or more activities of varying intensity with resulting average data reflecting an intermediate intensity [21].

Epochs which contain more than one behaviour are classed as transitional events and are often excluded from analysis, owing to their potential to be misclassified [6]. Previous studies have used training and validation datasets with the training dataset usually containing mutually exclusive behaviour epochs, though it is unclear whether the validation datasets include transitional behaviour epochs [6,12]. In a real-time classification system, segregation of these transitional epochs is not possible, as there is no prior knowledge of the animals' current or future behaviour [7]. Therefore, analytics that can handle the transition between behaviour classes must be developed.

A simple approach using a moving window sampling technique with overlap between epochs has been proposed to alleviate some of the signal classification issues associated with the transition between behaviours [12,14,22,23]. To address this, Nielsen et al. [24] used a moving average of 3 or 5 seconds of the motion index or step count values obtained from the IceTag3D™ device. If the moving average at a particular time was greater than a predetermined threshold value (the value separating behaviour classes), then it was classified as walking; otherwise it was classed as standing. The simple approach has obvious application when dealing with univariate feature data with a simple threshold model but its application to classify behaviour states in sheep using ear acceleration data with multiple feature algorithms is yet to be evaluated.

If accelerometers are to be utilised in commercial operations for remote livestock health and welfare monitoring, there is a need to develop a system capable of real-time classification of behaviour that is in-sync with data acquisition. The accuracy of accelerometers to classify livestock behaviours in near real-time influences its utility for future research and application in livestock production systems.

The objective of this study was to apply a behaviour state classifier developed on "clean state" data to an example of live data, including transitional states, to replicate data acquisition in a commercial

scenario. This was achieved using a moving window algorithm. The aim was to determine the performance of this moving window algorithm to classify continuous accelerometer signals from ear, leg and collar mounted devices into four mutually exclusive sheep behaviour states.

## 2. Materials and Methods

### 2.1. Animals & Instrumentation

Data collected for analysis in Barwick et al. [3] was re-used to develop a moving window behaviour state classification algorithm. The original study was conducted on "Kirby", one of the University of New England's SMART Farms, Armidale, NSW, Australia (Longitude 151°35′40″E, Latitude 30°26′09″S). All animal experimental procedures were approved under the University of New England Animal Ethics Committee, AEC14-066.

Five Merino x Poll Dorset ewes were randomly selected for instrumentation from a group of ten ewes. The remaining five ewes were retained as companion animals. A GCDC X16-mini MEMS accelerometer (Gulf Coast Data Concepts, MS, USA) configured to collect signals at 12 Hz was attached simultaneously to three locations on each candidate sheep: a neck collar, the anterior side of the nearside front shin and the ventral side of the offside ear. The devices were $50 \times 25 \times 12$ mm in size and weighed 17.7 grams with orientations of the $X$, $Y$ and $Z$ axis being dorso-ventral, lateral and anterior-posterior, respectively. To minimise deployment scale and ensure accurate observation, only one animal was equipped with the instruments at a time. At each point of deployment a single animal was randomly selected, without replacement, and equipped with the three accelerometers devices. Three animals (one instrumented and two non-instrumented) were then released into a small adjacent paddock (80 m x 6 m) for visual observation over a period of approximately 2.5 h. This process was repeated with each of the five randomly selected sheep. Upon release, the movement of each instrumented animal was monitored and video recorded with observations classified as per Table 1. Each second of accelerometer data was annotated with a specific behaviour. When animals were out of visual sight, behaviour was labelled as 'unknown'.

**Table 1.** Descriptions of the four behaviour states monitored.

| Behaviour | Classification Description |
|---|---|
| Grazing | Grazing with head down or chewing with head up either stationary or moving. No rumination included. |
| Walking | Minimum of 2 consecutive steps either forward/ back or sideways. |
| Standing | Stationary standing with minor limb and head movements. Animal is in a standing posture whilst idle or inactive. Head may be up or down. Rumination bouts included. |
| Lying | Animal is in a lying posture whilst idle or inactive assuming a recumbent position with minor head movements. Rumination bouts included. |

### 2.2. Developing the 'Clean State' Behaviour Classification Model

The same 'clean state' behaviour classification models developed by Barwick et al. [3] were used in the current study in their respective deployment locations. In brief, the behaviour-annotated accelerometer files were divided into 10 second epochs with unknown and transitional behaviour epochs (based on a visual assessment of the video recorded data) being excluded from the analysis. For each 10 second epoch, fourteen movement features were calculated from the raw acceleration measurements. The fourteen features were included in a random forest (RF) model for the purposes of ranking their importance in most closely estimating behavior. Feature importance ranking varies across deployment types as the mode of data collection yields different signals (i.e. the swing observed from an ear tag is not observed from the leg brace), and thus different features are better able to discriminate between behaviours. For each deployment method, the three calculated features with greatest importance were subsequently used in a quadratic discriminant analysis (QDA) classifier. As reported by Barwick et al. [3], the following QDA model feature combinations were identified as

yielding the highest prediction accuracies in their respective modes of deployment: collar (*Az*, *Entropy* and *AI*); leg (*Ax*, *SMA* and *MV*) and; ear (*MV*, *AI* and *Ay*). These features, which out of the initial fourteen features were considered relevant and were calculated, are listed in Table 2.

**Table 2.** Equations used to calculate features from raw *X*, *Y* and *Z* acceleration values.

| Feature | Equation |
|---------|----------|
| *Average X-axis (Ax)* | $A_x = \frac{1}{T} \sum\limits_{t=1}^{T} x(t)$ |
| *Average Y-axis (Ay)* | $A_y = \frac{1}{T} \sum\limits_{t=1}^{T} y(t)$ |
| *Average Z-axis (Az)* | $A_z = \frac{1}{T} \sum\limits_{t=1}^{T} z(t)$ |
| *Movement Variation (MV)* | $MV = \frac{1}{T} \left( \sum\limits_{i=1}^{T-1} \lvert x_{i+1} - x_i \rvert + \sum\limits_{i=1}^{T-1} \lvert y_{i+1} - y_i \rvert + \sum\limits_{i=1}^{T-1} \lvert z_{i+1} - z_i \rvert \right)$ |
| *Signal Magnitude Area (SMA)* | $SMA = \frac{1}{T} \left( \sum\limits_{t=1}^{T} \lvert a_x(t) \rvert + \sum\limits_{t=1}^{T} \lvert a_y(t) \rvert + \sum\limits_{t=1}^{T} \lvert a_z(t) \rvert \right)$ |
| *Average Intensity (AI)* | $AI = \frac{1}{T} \left( \sum\limits_{t=1}^{T} MI(t) \right)$ <br> Where $MI(t) = \sqrt{a_x(t)^2 + a_y(t)^2 + a_z(t)^2}$ |
| *Entropy* | $S = \frac{1}{n} \sum (1 + Ts_i) \ln(1 + Ts_i)$ <br> Where *n* is the number of records in the burst and *Ts* = *Ax* + *Ay* + *Az* |

Where T = time (calculated as the number of time samples in the epoch).

### 2.3. Application of the 'Clean State' Model to a Moving Window Behaviour Classification of Live Data

The performance of the 'clean state' algorithm was evaluated on a continuous accelerometer data stream including transitional behaviours and behavioural events shorter than 10 seconds. The data sets for each sheep within each deployment were fully annotated and contained a continuous stream of data across the entire observation period. The total duration of each dataset collected is shown in Table 3 (in Results and Discussion). The three features identified by RF for variable importance were calculated on the collar, leg and ear data sets for each individual sheep. Using the behaviour prediction model developed by Barwick et al. [3], activity for each sheep was classified using a 3, 5 and 10 second moving window epoch. The three moving window lengths selected (3, 5 and 10 seconds) were based on previously-used epoch sizes applied to group accelerometer data [6,12]. Raw accelerometer data was recorded at 12Hz, giving 12 rows of raw accelerometer data for each second, where each row represents one twelfth of a second. Moving window epochs of 3, 5 and 10 seconds were hence made up of 36, 60 and 120 signal rows, respectively. For each window, the window was stepped by 1 row and used an overlap of post behaviour signals to classify behaviour at a specific point. In the case of the three second moving window, row 1 was classified using signal row 1 plus the following 35 signal rows, equaling 36 rows. Row 2 used signal values from rows 2–37 and so on, where row *n* used signal rows *n* to (*n* + 36) and the value of *n* increases by 1 each time. Likewise, row 1 was classified using the 5 second window as rows 1–60, and the 10 second window as rows 1–120.

The data analysis workflow is summarised in Figure 1. Processing and analysis was conducted in Matlab (Mathworks, R2014a) and R (v3.1.2; R core Team, 2014).

To quantify the performance of the moving window classification algorithm, the following equation was used to calculate the total correct classification rate (θ):

$$\theta = \frac{number\ of\ model\ predicted\ behaviour\ events}{total\ number\ of\ annotated\ behaviour\ events} \times 100 \tag{1}$$

Here, $\theta$ denotes the percentage of points correctly predicted (in agreement with the video observations). These classification rates were calculated separately for each sheep within each deployment.

### 2.4. Determining the Most Suitable Moving Window Length

To test for a difference between the three moving window lengths on the accuracy of predicting each of the four behaviours (response variable), a linear mixed model (lmm) was developed. There was a total of four behaviours, three window lengths and up to five sheep included in the model (depending on the deployment mode). The number of sheep and behavioural observations differed between deployments (Table 3 in Results and Discussion). Deployment mode, window size and their interaction effect were set as fixed effects and Sheep ID was the random component. Post-hoc comparisons of least square means were achieved by using the Tukey procedure [25].

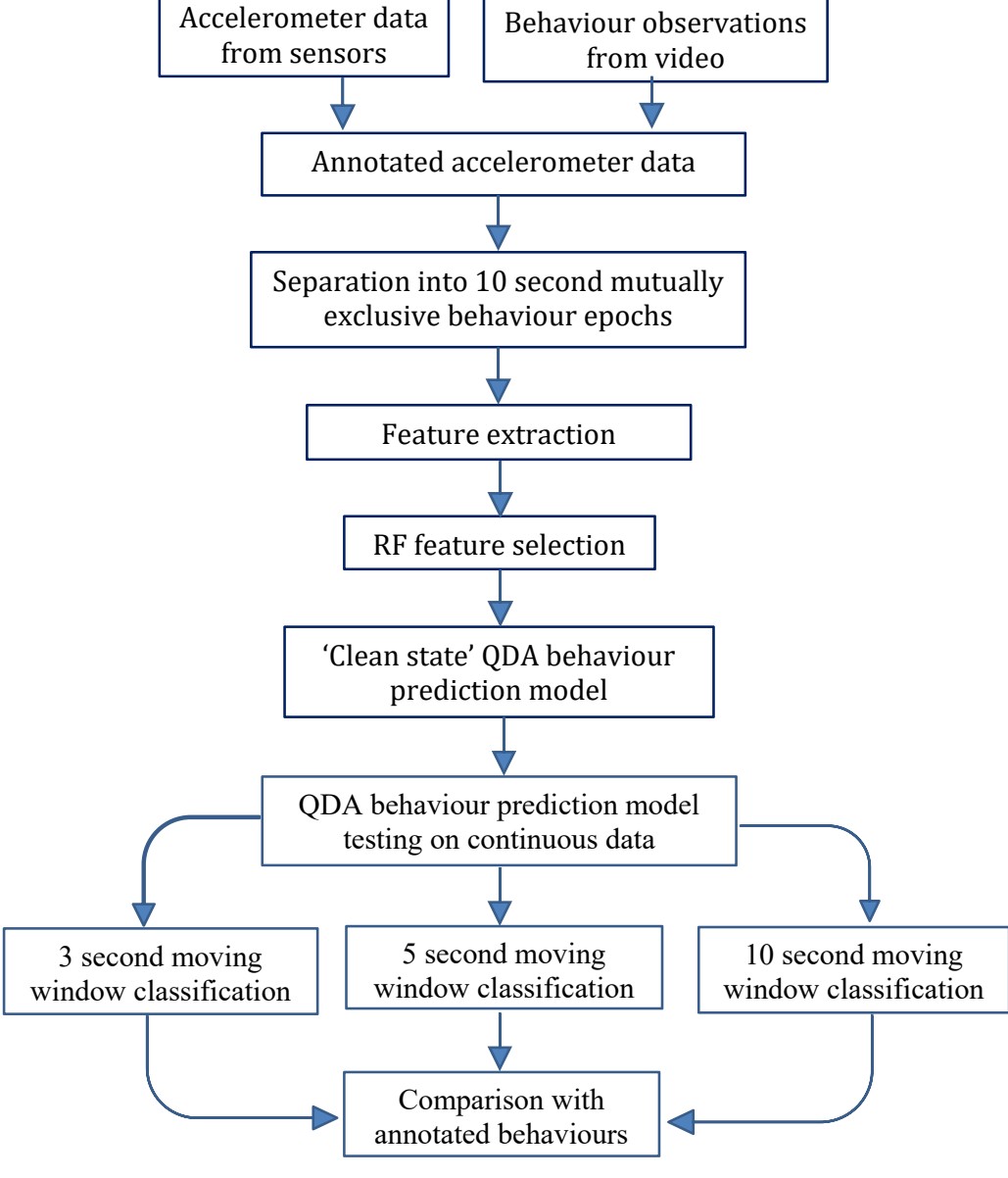

**Figure 1.** Schematic workflow for testing the behaviour prediction model across a continuous accelerometer signal stream using a 3, 5 and 10 second moving window.

## 3. Results and Discussion

There were no apparent adverse effects of sensor attachment on animal behaviour. A summary of the quantities of data collected for each state across the three deployment modes is given in Table 3. Observational periods were the same for all deployments as accelerometers were attached to the body of a given sheep in the three locations simultaneously. However, due to sensor malfunction, data quantities varied between deployments. In periods where limited data was collected because of sensor malfunction, a subsequent deployment was performed. Limited lying behaviour was recorded because of a lack of animal motivation to rest in a recumbent posture; therefore these results should be interpreted with caution.

**Table 3.** Duration (in minutes) of data collected for each deployment. Highlighted cells indicate no data was collected for those events.

| Sheep ID | Ear | | | | Leg | | | | Collar | | | |
|---|---|---|---|---|---|---|---|---|---|---|---|---|
| | Standing | Walking | Grazing | Lying | Standing | Walking | Grazing | Lying | Standing | Walking | Grazing | Lying |
| A | 22 | 14 | 21 | 0 | 9 | 7 | 14 | 9 | 19 | 15 | 30 | 7 |
| B | 55 | 10 | 4 | 0 | 7 | 11 | 14 | 15 | 8 | 12 | 15 | 16 |
| C | 35 | 17 | 14 | 0 | 0 | 6 | 9 | 0 | 0 | 0 | 10 | 0 |
| D | 56 | 10 | 5 | 0 | 0 | 0 | 0 | 0 | 0 | 0 | 0 | 0 |
| E | 10 | 3 | 23 | 0 | 9 | 4 | 22 | 19 | 10 | 4 | 23 | 20 |
| Total minutes | 178 | 54 | 67 | 0 | 25 | 28 | 59 | 43 | 37 | 31 | 78 | 43 |

In comparison with video observations, there were considerable differences in the classification performance across the four behaviours and between animals within each deployment (Table 4). Deployment mode also had an influence on the between-animal classification accuracies, with the leg and collar deployments showing a greater range in successful classification accuracies compared to the ear attachment, especially for standing behaviour. This is consistent with the original classification algorithm performance and the lower prediction accuracies for standing behaviour in these modes of attachment [3]. There were significant differences observed in the prediction accuracy of standing behaviour between the ear and leg deployments using a 10 second moving window (d.f. = 24, t.ratio = 2.515, $P < 0.05$) and for walking behaviour between the collar and leg (d.f. = 27, t.ratio = −2.559, $P < 0.05$) and collar and ear (d.f. = 27, t.ratio = −3.033, $P < 0.05$) deployments using the 3 second moving window classifier. There were no other significant ($P > 0.05$) differences observed for the interaction between window length and deployment mode for the remaining combinations.

**Table 4.** The percentage of correctly predicted behaviours for individual sheep in each deployment mode across the three moving window lengths evaluated (3, 5 and 10 seconds). Percentage values derived from agreement between visual annotation and classification algorithm prediction. Greyed areas indicate where data was not collected.

| Deployment | Sheep ID | 3 s Window | | | | 5 s Window | | | | 10 s Window | | | |
|---|---|---|---|---|---|---|---|---|---|---|---|---|---|
| | | Standing | Walking | Grazing | Lying | Standing | Walking | Grazing | Lying | Standing | Walking | Grazing | Lying |
| Leg | A | 55% | 95% | 74% | 95% | 54% | 95% | 78% | 95% | 51% | 94% | 84% | 94% |
| | B | 1% | 92% | 83% | 36% | 1% | 92% | 86% | 27% | 0% | 91% | 90% | 27% |
| | C | | | 64% | | | | 66% | | | | 73% | |
| | E | 79% | 93% | 78% | 13% | 78% | 96% | 83% | 0% | 79% | 93% | 78% | 13% |
| | Mean prediction % | 48% | 93% | 76% | 38% | 48% | 94% | 80% | 29% | 47% | 94% | 81% | 35% |
| Collar | A | 67% | 65% | 85% | 12% | 67% | 68% | 85% | 11% | 62% | 74% | 84% | 9% |
| | B | 30% | 80% | 85% | 4% | 29% | 84% | 84% | 4% | 27% | 87% | 85% | 4% |
| | C | 76% | 95% | 92% | | 67% | 95% | 92% | | 24% | 96% | 93% | |
| | E | 96% | 43% | 94% | 5% | 96% | 47% | 94% | 5% | 95% | 61% | 94% | 5% |
| | Mean prediction % | 68% | 87% | 88% | 6% | 67% | 88% | 88% | 6% | 64% | 90% | 88% | 6% |
| Ear | A | 80% | 89% | 85% | | 78% | 92% | 86% | | 80% | 94% | 87% | |
| | B | 97% | 98% | 55% | | 97% | 99% | 62% | | 97% | 100% | 58% | |
| | C | 69% | 86% | 79% | | 70% | 86% | 84% | | 69% | 85% | 89% | |
| | D | 96% | 100% | 76% | | 96% | 100% | 84% | | 97% | 100% | 92% | |
| | E | 87% | 99% | 89% | | 86% | 100% | 92% | | 87% | 100% | 95% | |
| | Mean prediction % | 89% | 92% | 83% | | 89% | 93% | 86% | | 89% | 93% | 89% | |

The accuracies across the three moving window sizes varied little within each deployment mode. One of the main disadvantages of using a longer window is that the window may contain a mixture of two or more activities of varying intensity and the average data may reflect an intermediate intensity. For example, if a sheep's activity abruptly changes from resting to walking, the intermediate moving average may be incorrectly ascribed to a period of eating. Additionally, if the activity bout is shorter than the window, the average value for that burst will differ from the actual activity intensity, again leading to misclassification. A longer window has the advantage of normal data-smoothing through time averaging and this is evident in the classification of walking behaviour from the collar deployment, with the longer window length having higher classification accuracy (although this difference is not statistically significant P > 0.05, d.f = 27, t.ratio = −0.636). This is further supported by the longer window lengths having fewer behaviour transitions in comparison to the 3 and 5 second window predicted data files.

There was a much larger number of behaviour transitions in the predicted datasets than was apparent in the annotated data file (Table 5). This is a result of the moving window algorithm classification process, as each row of data could ultimately be given a different behaviour prediction than the preceding row. This presents a challenge for the current classification methodology because of the reasons outlined with varying epoch lengths. An approach worthy of future research may be to use exclusive epoch segments with no overlap which will reduce the number of behaviour transitions within the predicted dataset, more closely aligning to that seen in video recordings. Consequently, and in accordance with Smith et al. [7], we found that window selection becomes a pragmatic trade-off to ensure the moving windows are sufficiently long enough to represent behaviours, while being short enough to reduce the likelihood of multiple behaviours being captured in the same window.

**Table 5.** Total number of behaviour transitions in the annotated and predicted data sets for individual sheep in each deployment mode across the three moving window lengths. Greyed areas indicate where data was not collected.

| Animal ID | Window Length | Ear | | Collar | | Leg | |
|---|---|---|---|---|---|---|---|
| | | Annotated | Predicted | Annotated | Predicted | Annotated | Predicted |
| A | 3 s | 34 | 415 | 49 | 2196 | 50 | 309 |
| A | 5 s | 34 | 263 | 49 | 1488 | 50 | 206 |
| A | 10 s | 34 | 130 | 49 | 803 | 50 | 101 |
| B | 3 s | 3 | 233 | 106 | 1502 | 106 | 396 |
| B | 5 s | 3 | 149 | 106 | 1009 | 106 | 240 |
| B | 10 s | 3 | 79 | 106 | 560 | 106 | 94 |
| C | 3 s | 116 | 756 | 2 | 345 | 2 | 126 |
| C | 5 s | 116 | 496 | 2 | 226 | 2 | 103 |
| C | 10 s | 116 | 271 | 2 | 114 | 2 | 53 |
| D | 3 s | 3 | 326 | | | | |
| D | 5 s | 3 | 156 | | | | |
| D | 10 s | 3 | 81 | | | | |
| E | 3 s | 69 | 561 | 89 | 573 | 90 | 421 |
| E | 5 s | 69 | 259 | 89 | 436 | 90 | 227 |
| E | 10 s | 69 | 114 | 89 | 328 | 90 | 421 |

A behaviour transition is defined as the period in which the animal's behaviour changes from one state to another, e.g., standing to walking. The number of behavioural transitions varied considerably between the annotated and predicted datasets. These decreased in the predicted datasets as the moving window length increased which may explain the slight improvements in classification success for the 10

second window. This can be explained given the number of transitions in the 10 second window length predicted dataset being closer to the actual number of behaviour transitions in the annotated dataset.

*3.1. Leg*

Window length was shown to have no significant (P > 0.05) effect on the behaviour prediction in any of the four behaviours recorded for the leg deployment. There was little variation across the three window lengths for the prediction of walking, with all three moving window lengths achieving greater than 90% accuracy for individual sheep. Similar results were reported by Nielsen et al. [24] in cattle, with an average of 10% misclassification rates in walking periods using the IceTag3D® device. Standing prediction accuracies were consistent across the three window lengths in this study, however there was large variation in prediction accuracy between sheep (0% to 79%). During model development, standing behaviour was often misclassified as grazing. Therefore, this misclassification is also evident in the moving window classifier used in this current study.

In terms of limb movement, grazing is characterised by infrequent steps followed by periods of minimal leg movement. When an animal is grazing, a leg attached accelerometer only detects the dynamic acceleration associated with the locomotive movements (i.e., steps taken in search of new pasture). The support phases of locomotion are extended during grazing activity and hence, the accelerometer is often recording static acceleration. This results in similar acceleration signals between grazing and standing behaviour (created by the extended support phase). Therefore, standing events were often misclassified with grazing using the current QDA model [3]. This presents a challenge for the prediction of behaviours from leg mounted sensor, particularly behaviours that share similar dynamic leg movement. Prediction success may differ in response to the animals grazing environment as this can influence the speed at which animals search for new areas of available pasture, i.e., high levels of pasture may result in a higher level of misclassification of standing with grazing as animals move at a slower rate in comparison to short pastures where the animals have a higher level of movement as they seek out pasture.

There was no significant (P > 0.05) difference between the three moving window lengths for the prediction of grazing behaviour. However, on average, grazing behaviour was best classified by the 10 second moving window with between-sheep values ranging from 73% to 90% (Table 4). Lying behaviour was well predicted for sheep A with the 3, 5 and 10 second windows recording accuracies of 95%, 95% and 94%, respectively. This is reflective of the clear differentiation between upright and lying postures based on the static acceleration values in relation to the gravitational field, when attached to the leg. Interestingly, sheep B and E had low prediction accuracy for grazing behaviour. This is likely associated with sensor movement on the leg for these two sheep resulting in a change in axis orientation, affecting the recorded acceleration for the Average X-axis feature which was used in the leg QDA algorithm. This highlights the importance of having metrics that use absolute acceleration, reducing the impact of sensor orientation on classification success.

*3.2. Collar*

Window length was found to have no significant (P > 0.05) effect on the accuracy of behaviour prediction due to the four behaviours recorded for the collar deployment. The average classification accuracy for walking improved after lengthening the window from 3 to 10 seconds, although not significantly. However, there was substantial variation in classification success between sheep with accuracies ranging from 61% to 96%. This between-animal variation with collar deployed sensors has previously been described by Trotter et al. [8] and Hämäläinen et al. [26], and is due to variation in sensor placement between individuals. Collars can move freely around the animal's neck, thus the position of the accelerometer may not always be the same which may give inconsistent readings despite the posture of an animal remaining exactly the same. It has been reported that even a slight deviation in sensor orientation can influence the classification results [27].

Classification success rate for the current study was higher than that reported for cattle by González et al. [4] (~60%). These authors found travelling events were commonly misclassified with foraging behaviour. This discrepancy may be due to differences between the either travelling speed of sheep and cattle or the dynamic motion of collar movement. As sheep are physically smaller, their step length is shorter whilst step rate may be greater. Therefore, accelerations could be amplified due to greater dynamic sensor movement resulting in stronger acceleration signals and better analytical performance. In the current study, standing was consistently classified across each moving window, again with substantial difference between sheep being observed with values ranging from 24% to 96%. Grazing behaviour was also consistently classified across the three moving windows with high sensitivity (88%). Lying behaviour was very poorly predicted from the collar data, with a sensitivity of 6% for all moving windows. The small amount of data used for this behaviour in model training more than likely contributed to this poor classification, along with similarities in the sensor output between lying and other behaviours. Further research is warranted to explore this issue of data imbalance.

### 3.3. Ear

There was no significant ($P > 0.05$) effect due to window length on behaviour prediction of any of the three behaviours recorded in the ear deployment. The moving window length of 10 seconds yielded slightly better mean prediction accuracies compared to the 3 and 5 second window lengths although this difference was not significant.

The mean prediction percentages for the ear were the highest for all three deployment modes, a function of the clear differentiation between raw acceleration signals across grazing, standing and walking behaviours. Between-sheep variation was evident in the prediction of standing and grazing behaviour and to a lesser extent in walking activity. Increasing the moving window from 3 to 10 seconds yielded a small improvement in classification agreement, particularly for grazing behaviour which achieved an average 6% improvement with the longer window. Walking behaviour was consistently predicted, yielding a between-sheep accuracy range of only 15%. For the 10 second moving window, standing behaviour prediction accuracies ranged from 69% to 97%, walking prediction accuracies ranged from 85% to 100%, and grazing behaviour predictions ranged from 58% to 95%. For the 5 second moving window, standing behaviour prediction accuracies ranged from 70% to 97%, walking prediction accuracies ranged from 86% to 100%, and grazing behaviour predictions ranged from 62% to 92%. For the 3 second moving window, standing behaviour prediction accuracies ranged from 69% to 97%, walking prediction accuracies ranged from 86% to 100%, and grazing behaviour predictions ranged from 55% to 89%. The larger variation in grazing predictions is also reflected in the original classification model showing misclassification with standing and walking behaviours. This can be explained by the static acceleration signatures produced within these two behaviours. As an animal's head is lowered to the ground to graze, the sensor's freedom to hang sees it maintain a similar orientation to that during standing and walking. This problem is relevant when using the values associated with a particular axis (e.g., average X or average Y) which are used in the clean state model tested here. The variation in sensor orientation between animals is an issue with the ear-tag form factor arising from the susceptibility of the tag to rotate in the ear due to the 'single pin' method of fixation. Furthermore, the physical differences in ear structures between animals can influence the dynamic motion experienced by the sensor affecting the acceleration signals recorded across behaviours.

## 4. General Discussion

For on-animal behaviour monitoring systems to function commercially, the data must be captured, processed and analysed in sync with data acquisition. This is inherently more challenging than an offline classifier as in a forward propagating sense, the behaviour state model will be challenged by the transition between behaviours as it can only look at historic data, for example the previous 60 seconds of data to make a behaviour prediction. Therefore, models have to be capable of operating in a manner that can determine when behaviour state changes across a string of signals containing multiple behaviours.

Furthermore, models must be able to account for between-animal variations in signal output. This is shown in the present study by the substantial classification differences between animals within the same behaviour category. This is particularly evident for standing behaviour predicted from the collar and leg deployments. Similar results were found by Blomberg [28], who reported accelerometers correctly classified behaviour in cattle with a relatively high rate of accuracy, but with large differences between individuals. While this current study did not investigate the specific causes of variability between sheep, it is speculated that the primary source of this between-animal variation in signals arose from two related causes. Firstly, animal physical characteristics affecting how the sensor attaches to the animal and secondly, the sensor motion during activity. Wolfger et al. [29] speculated that the inter-animal variation between estimates provided by observations and accelerometer recordings could be attributed to differences in ear movement between cattle. Similarly, differences between sheep in physical (skeletal/muscle/tendon) structure may influence their walking pattern (or gait) and thus sensor motion during measurement of activity. Consequently, the models' ability to discriminate between behaviour states may be adversely affected. González et al. [4] also observed large differences between animals with collars for the fitted parameters of the probability density functions, threshold values and structure of the frequency distributions (e.g., overlap of populations). Further research should investigate the source of such variation to determine the proportion of observed variation between experimental animals that result from differences in animal movement patterns, sensor attachment and measurement between sensors [4], and whether or not it can be negated through the use of larger model training datasets or even development of models that accommodate or adapt to individual animals. This has huge implications for commercial adoption of this technology, as sensors will be deployed across a variety of animals of different age, breed, production status, and so on, and prediction algorithms need to account for this between-animal variation in their classification protocols.

A potential limitation of this current study is the small amount of data used for model development. It is important that enough data is recorded in order for the behaviour model to capture the variation between animals and also the variation within behaviour signals of an individual animal. A much larger cohort of test subjects is required for extensive validation of this classification approach. Furthermore, the data used for model development in Barwick et al. [3] was unbalanced which can lead to the model overfitting certain behaviours, with small changes in the training dataset resulting in different behaviour predictions. To overcome this, up or under-sampling to equalize the data for each behaviour category in the training dataset can be used. Sakai et al. [11] compared the precision and sensitivity of behaviour classification before and after balancing, reporting mixed results. A similar approach may be warranted with the current study to determine the effect of balancing data on the classification performance of the testing dataset.

## 5. Conclusions

The continuous monitoring of sheep behaviour with on-animal sensor devices presents many potential opportunities with respect to more precise management of animal health and welfare. This current study tested a 'clean state' behaviour model on a forward propagating stream of data that emulated a live data feed to simulate a commercially deployed data acquisition. Results show that an ear-borne tri-axial accelerometer was the superior mode of deployment to discriminate behaviour in sheep using a moving QDA algorithm developed from 'clean state' data. No significant difference was observed between the three moving window lengths evaluated. The collar and leg modes of deployment yielded large variations in prediction accuracy between behaviours given the similarities in acceleration signal across activity classes. Coupled with this varying ability to classify behaviour, both the leg and collar attachments present practicality issues with commercial deployment. Furthermore, with the inclusion of transitional behaviours in this live data stream simulation, a large between-sheep variation in classification rate was found. This must be considered in future work, highlighting the need to obtain a large dataset for algorithm development across multiple environments and sheep classes and ensuring standardisation of how the devices are attached to the animal. Addressing simpler

modelling challenges, i.e., modelling which just focusses on active and inactive states, has been shown to average out the effect of individual animal peculiarities on prediction accuracy. More research is required to ascertain the relevance of fine scale behaviour delineation.

**Author Contributions:** Conceptualization, J.B., D.W.L., R.D., M.W., D.S. and M.T.; methodology, J.B., R.D., M.W., D.S. and M.T.; software, M.W. and D.S.; formal analysis, J.B., R.D., M.W. and D.S.; data curation, J.B. and D.S.; writing—original draft preparation, J.B.; writing—review and editing, J.B., D.W.L., R.D., M.W., D.S. and M.T.; funding acquisition, J.B. and M.T. All authors have read and agreed to the published version of the manuscript.

**Funding:** This research was funded by the Sheep Cooperative Research Centre (CRC), the University of New England School of Science & Technology, and the Commonwealth through an Australian Postgraduate Award.

**Acknowledgments:** One of the Authors (David W. Lamb) would like to acknowledge the support of Food Agility CRC Ltd., funded under the Commonwealth Government CRC Program. The CRC program supports industry-led collaboration between industry, researchers and the community.

**Conflicts of Interest:** The authors declare no conflict of interest.

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
