# Peer review of "Identifying Sheep Activity from Tri-Axial Acceleration Signals Using a Moving Window Classification Model"

_remotesensing, doi:10.3390/rs12040646_

Round 1

Reviewer 1 Report

The followings are the points which should be revised:

Abstract

L14 Change “Sheep behavior” to “Behavior”, since the content of these sentences for introduction is applied also for other animal species.

L32-33 This is true, but there might be other sources of such variations such as differences in movement patterns among individuals. So, please rewrite the sentence to clarify that sensor orientation is “one of the sources of the variations”.

Keywords

Rearrange them as an alphabetical order.

Introduction

L46 Rewrite as “established in ruminants (cattle [4-8], goats [9, 16, 17] and sheep [3, 10-15])”.

L53 “the commercial” to “commercial”

L84 I can understand the meaning of “a predetermined clear value”, but the meaning of “clear value” might be unclear for readers. Should add the explanation.

Materials and Methods

L146 Change “in Table 3” to “in Table 3 (in Results and Discussion)”.

L173 As well, change “Table 3” to “Table 3 (in Results and Discussion)”. Rewrite the sentence from “Deployment mode ~” to “Deployment mode, window size and their interaction effect were set as fixed effects”.

L175 Delete “with an interaction between Window and Deployment”. The interaction effect has to be included in the linear model and lsmeans is usually just for estimating least squares means for the effect. Besides, add the type of method used for the statistical comparison (is Tukey?).

Results and Discussion (change “&” to “and”)

L235 “(see Table 5)” to “(Table 5)”

L237 Change these three p values to “P<0.05”.

L251 “not significant” to “not statistically significant”

L272 “A approach” to “An approach”

L319 “Table xx” is “Table 4”?

L368-375 From “For the 10 second” to “55% to 89%”, it’s not necessary to show the results in the long sentences, because these results are shown in Table 4. Delete or simplify the sentences.

General discussion

L422 Delete the sentence “The current study does account for this using five test animals.” and revise “However, a much” to “A much”, because the data for “lying” in the present study is insufficient for clarifying the variation within behavior signals of an individual animal. Although the authors indicated in the revision letter that “the amount of data used for this study is comparable to ~, for example Alvarenga et al. (2016)”, Alvarenga et al. conducted the data collection twice (four hours for each) to collect more data on lying and standing.

L430-431 “to determine the effect of balancing the data on the classification ~”

References

Recheck and reformat the style of the references. All reference has to be formatted according to the instruction of the journal and other papers in the journal, such as:

Fogarty, E.S.; Swain, D.L.; Cronin, G.; Trotter, M. Autonomous on-animal sensors in sheep research: A systematic review. Comput. Electron. Agric.(in Italic) 2018 (in Bold), 150 (in Italic), 245-256. (and add the doi if available)

Reviewer 2 Report

Review remotesensing-715100

General comments

The present manuscript studies the possibility to use animal-borne acceleration sensors to classify animal behavior using moving window approach. The authors tested three different sensor deployments (ear, leg, collar) and three different window length to evaluate the behavior of five sheep. The study is interesting and provides a relevant contribution to develop methodology to monitoring the welfare of farmed animals. Yet, I had some problems to understand throughout the approach and the statistical analyses the authors used, and recommend the authors to clarify those. I also suggest the authors to double-check the values of mean prediction values given in Table 4.  

Detail comments

Line14: Accounts for all animal, please remove ”sheep”

Line 18-21: long sentence that is difficult to follow, please rephrase

Line 135-137: not whether I understand this correctly – does it mean that, depending on deplotment (ear, leg, collar), you used different equations to define the same features and thus behaviors? Please clarify, and why you did so

Line 150-152: should the number of rows involved not differ among the three moving window (3,5, or 10 sec)? I am sorry, but I have problem to understand this approach. Please clarify

Line 169-175: what is your response variable? Behavior per row? Accuracy of predicting per row? Did you run one model per behavior? Have you tested whether you may need to add a correlation structure, rows closer in time might be more similar to each other than those more apart? Please consider to add a table showing the model structure to clarify your approach

Line 237: please add information on df, F or t stats, next to the p-value

Line 249-251: if it is not significant than there is no difference. Please add information on p-value, df, t-value or F stats, etc

Line 319: please add the table number

Line 323-327: why would that account for two sheep, but not for the others? Please clarify

Line 367: in line 360 you write that difference are not significant, thus, there is not improvement with a longer window

Line 372: please add space between from86%

Table 4: please double-check the mean prediction % throughout the table, e,g., (55+1+79)/3=45, not 48.

Table 5: includes redundant information to Table 4, please merge the two tables and remove table 5

Round 2

Reviewer 2 Report

I am grateful to the authors for their effort in revising their manuscript according the comments made and clarifying their approach. I am fine with all changes made, except that I still think that df, Fstats etc would be interesting to include and that the manuscript would benefit from merging Table 4 and 5 in order to condense it. I think Table 4 does not add so much information to the manuscript. Sheep/behavior with NAs are already given in Table 3. To show the variation among sheep, stating values in Table 5 with +/- SD and/or min and max values would do fine as well.  

Author Response

Point 1: I am grateful to the authors for their effort in revising their manuscript according the comments made and clarifying their approach. I am fine with all changes made, except that I still think that df, Fstats etc would be interesting to include and that the manuscript would benefit from merging Table 4 and 5 in order to condense it. I think Table 4 does not add so much information to the manuscript. Sheep/behavior with NAs are already given in Table 3. To show the variation among sheep, stating values in Table 5 with +/- SD and/or min and max values would do fine as well.

Response 1: Thank you to the reviewer for their time and feedback.

We have added the df and t ratio values in the appropriate areas throughout the manuscript.

Table 5 has been deleted and the mean prediction value for each behaviour within the three deployment modes across the three moving window lengths have been added to table 4. Table 4 is valuable for this article as it shows the between animal variation in behaviour prediction within each mode of deployment. The duplication of information has been removed with the deletion of Table 5. The additional information which table 5 contained can be viewed in the referenced paper (Barwick et al. 2018).

This manuscript is a resubmission of an earlier submission. The following is a list of the peer review reports and author responses from that submission.

Round 1

Reviewer 1 Report

This paper focuses on the effects of differences in moving window length (3, 5 and 10 seconds) and mount position of accelerometer (ear, collar and leg) on the performance of behavioral classification for sheep. The results of the manuscript show that, although the effects of difference in mounting position and individuals on the performance were observed, no significant difference in model accuracy was observed between the three window lengths. Consequently, the authors concluded that a moving window classifier might be capable of using continuous accelerometer signals for near-real-time behavioral classification and indicated the possibility to use this algorithm for remote behavioral classification using acceleration data in near future. The topic might be within the scope of the journal and this manuscript provides some methodological information on the use of on-site acceleration data in animal production systems. However, as yet this manuscript this manuscript requires revisions to meet the standard of the journal, based on the following:

The major concern for this manuscript is the interpretation of the results by the linear mixed model the authors used. The authors indicated in “2.4. Determining the most suitable moving window length” that, the accuracy of prediction was statistically analyzed with a linear mixed model including the fixed effects of deployment mode and window length and the random effect of animals. But there are no tables which show the results for the ANOVA, and the authors only indicated that “there are no significant effects of window size” for “each deployment mode” from the discussion chapter “3.1~3.3”. If the authors want to say the effects of the difference in window size for each deployment mode, it is necessary to include the interaction effect of the two fixed effects and to show the result. Showing results of the analysis by a linear mixed model, with inclusion of the interaction effect, is needed to indicate clearly the effects of window size and deployment mode on the prediction accuracy. Otherwise, readers cannot understand whether the mounting position significantly affected the accuracy or not, and whether the effect of window size on the accuracy could be changed according to the difference in mounting position of the sensor or not. In addition, although the authors mentioned no significant effects of window size, it’s written that increasing the window length could improve the accuracy. It seems to be contradictory result. When the result of statistical analysis show “no significant effects”, authors should not mention the difference as a meaningful difference. So, rewrite the discussion section “3.1~3.3” and other parts in the manuscript in which the authors indicate the improvement of accuracy by prolonging the window size.

The second question is about the data of behavior transitions. With Table 6, the authors indicated “the number of transitions decreased in the predicted datasets as the moving window length increased which may explain the slight improvements in classification success for the 10 second window”. But I cannot understand the result. In my feeling, when window length increases, overlap parts of behavior also increase and longer windows should cover many different behaviors. Is not correct? To clarify the result, add more explanation about the way to treat behavior transitions in details and show not only the total number but also the proportions of the number of behavior transitions to total number of data for each animal, window length and deployment mode in Table 6.

The followings are other comments which should be revised:

L47 There are also some studies which reported the capability of accelerometers to evaluate activity of goats, such as Zobel et al. (2015: J. Dairy Sci. 98 :1082–1089) and Sakai et al. (2019: Comput. Electron. Agric.166: 105027).

L79 and other parts The authors should revise the way to put citations in the manuscript; for example, “To address this, [23] used” should be “To address this, Nielsen et al. [23] used” or you should change the sentence in order to put the citation number at the end of the sentence. The writings such as “by [23]” and “in [3]” are used in many parts of the manuscript, but such sentences should be also revised to put citation numbers without showing the authors’ names.

L103-106 It’s written that “~ was attached simultaneously to three locations on each candidate sheep”. Then I cannot understand why the duration of data collected for each deployment was different between the four behaviors in Table 3? Also for other tables, I cannot understand why the results for lying only for “Ear” were lacked? Please add the explanation in the text and in footnotes of the tables.

L112 Why the authors obtained the data of visual observation in such a short time (2.5h)? It’s quite a short period for building a model of behavioral classification. Please add the reason and/or explain the small number of data in a section as a “Limitation of this study”. Moreover, the four behaviors classified are quite unbalanced, and the unbalanced data with a small size of data might be a cause of classification problem. Not only in other fields such as informatics, but also in animal science some studies focused on the problem of unbalanced data for behavioral classification (e.g. Williams et al. (2016: J. Dairy Sci. 99:2063–2075), Sakai et al. (2019: Comput. Electron. Agric.166: 105027)). So, in this chapter or in Discussion section, please add some implications for the problem in behavioral classification. Moreover, as for references, some studies which might be related to this study should be cited, such as Walton et al. (2018: R. Soc. open sci. 5: 171442).

L120 and others The number of Tables should be ordered according to the appearance in the manuscript. “Table 5” in this sentence should be “Table 2”, and change the order of Tables also for other tables.

L129 Why only seven features were used for RF? As indicated in the manuscript, Ax, Ay and Az are orientation-dependent features and I think other features should be included in order to minimize the effect of orientation of accelerometer on model performance. Please add the explanation to choose the features, and if possible, try to increase features and reanalyze.

Table 2 Add footnotes to explain the variables “T” and “N”.

L139-144 As commented in the general comments above, again, please add more explanation to clarify the relationship between window sizes and behavioral transitions.

L158 As indicated in the general comments, add the interaction effect between the fixed effects and reanalyze it.

Table 3 Which cells are highlighted? Please separate the line under “Ear”, “Leg” and “Collar” (between “Lying” and “Standing”). Why the data for Leg and Collar of the animal D were lacked? Please add the reason in the manuscript.

Table 5 How about other performance indicators? Add other indicators, such as precision and F1 score.

Table 4 Add lines under “3 sec window”, “5 sec window” and “10 sec window”.

Table 6 As in the general comments, add the percentage of the number of behavioral transitions to the total number of data for each window length.

P9L8-Conclusion (Do not change line numbers suddenly from a middle page) Rewrite the results and discussion in order not to emphasize the improvement of predictive performance with longer window length, since the effect of window size on the performance was not statistically significant.

Reviewer 2 Report

Please fine comments in the attached document
